# Correlations between Histopathological and Confocal Reflectance Microscopy Aspects in a Patient with Bowenoid Papulosis

**DOI:** 10.3390/diagnostics13091531

**Published:** 2023-04-24

**Authors:** Elena Codruta Cozma, Ana Maria Celarel, Ioana-Valentina Stoenica, Mihai Lupu, Laura Madalina Banciu, Vlad Mihai Voiculescu

**Affiliations:** 1Department of Pathophysiology, University of Medicine and Pharmacy of Craiova, 200349 Craiova, Romania; 2Department of Dermatology and Allergology, Elias University Emergency Hospital, 011461 Bucharest, Romania; 3Department of Dermatology, MEDAS Medical Center, 030447 Bucharest, Romania; 4Department of Dermatology, Carol Davila University of Medicine and Pharmacy, 050474 Bucharest, Romania

**Keywords:** bowenoid papulosis, confocal microscopy, non-invasive diagnosis, genital lesions, HPV

## Abstract

Bowenoid papulosis is a cutaneous disease that is part of the spectrum of genital in situ carcinomas, caused primarily by infection with oncogenic strains of the HPV virus. The potential to transform into squamous cell carcinoma requires the diagnosis and treatment of the lesions. We present the case of a 34-year-old non-smoker without medical history who presented to our clinic for the appearance of multiple, asymptomatic, well-defined, flat, pigmented violaceous papules at the root of the penis in evolution for a year. Reflectance confocal microscopy (RCM) suggested the diagnosis of bowenoid papulosis, which was confirmed by histopathological examination. The treatment with Imiquimod 5% (3 times/week) and Isoprinosine (4 g/day) was initiated, followed by monitoring of the lesions by repeated RCM examination. The evolution of the patient at 6 weeks of therapy was favourable, with clinical remission of lesions and improvement in RCM aspects of the evaluated skin. In conclusion, RCM represents a useful noninvasive examination method that allows not only the diagnosis but also the follow-up of the treatment response in order to decide the appropriate length of therapy.

## 1. Introduction

Bowenoid papulosis (BP), first described by Lloyd in 1970, then by Kopf and Bart in 1977, is a rare sexually transmitted disease, usually found in sexually active (30–40-year-old) patients and is considered, along with Bowen’s disease and erythroplasia of Queyrat, as a part of the histopathological spectrum of in situ genital carcinomas [1,2,3,4,5]. On the other hand, some authors consider BP a transitional form between genital condylomas and in situ squamous cell carcinoma [3]. 

Regarding the aetiology, infection with the oncogenic strains of Human Papilloma Virus (HPV) and especially infection with HPV 16 and 18 seem to be an important risk factor in the occurrence of BP, but also in the possible subsequent evolution towards squamous cell carcinoma (SCC) [6]. Clinically, BP can be characterised by hyperpigmented or skin-coloured papules and plaques with a slightly irregular surface, sometimes verrucous, with a diameter of less than 1 cm, frequently asymptomatic, located most often at the genital level, an aspect that sometimes causes difficulties in the differential diagnosis with condylomas genitalia [7]. Furthermore, histopathological examination revealed the presence of multiple atypical keratinocytes with hyperchromatic nuclei, with an increased amount of mitosis. Nevertheless, the evolution is in most cases favourable (with spontaneous regression in some cases), despite the histopathological background, which, in many cases, is difficult to differentiate from the one found in in situ squamous cell carcinoma. Therefore, clinical examination and dermoscopy are usually sufficient to guide the diagnosis and establish therapeutic management. At the same time, reflectance confocal microscopy (RCM) and biopsy with histopathological examination can be useful not only in establishing the diagnosis but also in following the evolution under treatment or establishing the nature of long-term unresponsive lesions. However, the invasive nature of the biopsy makes it difficult to use repeatedly during the treatment. On the other hand, the RCM allows re-evaluation of the lesion’s evolution whenever necessary, with minimal adverse effects on the patient [8].

## 2. Case Presentation

We present the case of a 34-year-old male presenting to the dermatology unit with asymptomatic lesions of approximately one-year duration overlying the base of the penis. Clinical examination revealed multiple, well-defined, flat, pigmented violaceous papules with velvety surfaces. The lesions were situated over the penile shaft at the base of the penis (Figure 1A). In addition, smaller erythematous, well-defined papules were found on the adjacent area. Genital warts, lichen planus pigmentosus and bowenoid papulosis, were suspected among the clinical diagnosis.

The lesions appeared one year ago and were accompanied by pruritus. Initially, the papules were small and brown in colour. In addition, the foreskin was described by the patient as being thin and dry. Simultaneously, the patient described pollakiuria, for which he was tested for a urinary tract infection and treated with systemic antibiotics.

From the patient’s medical history, we also noted a diagnosis of ankylosing spondylitis (since 2012), for which he received different treatments over time. At the time of presentation, the underlying disease had been in remission for more than 2 years, with no medication being administered at that moment. In addition, a diagnosis of seborrheic dermatitis was made in 2008, which was under control with topical treatments. 

In order to better assess the genital lesions, we performed a dermoscopy exam that revealed a pigmented papillomatous surface, brown-grey dots arranged at the periphery of the lesion and widespread dotted vessels suggesting bowenoid papulosis (Figure 1C).

An RCM examination was performed in order to obtain more information regarding these lesions, followed by a skin biopsy from one of the papules to confirm the diagnosis. The RCM exam revealed acanthosis, alteration of the honeycomb pattern, with the presence of multiple irregular bright cells in the dermal papillae corresponding to melanophages. The histopathology (Figure 2) exam revealed atypical keratinocytes throughout the entire thickness of the epithelium with increased mitotic activity, suprabasal mitoses, bowenoid nuclear atypia and focal melanic pigmentation with melanophages and dilated blood vessels present in the papillary dermis. The HPV genotyping exam was also performed but with negative results.

Given the clinical and histological features, the case was diagnosed as bowenoid papulosis. The patient was treated topically with Aldara 5% cream (Imiquimod) applied 3 times per week (for 6 weeks) and systemically with Isoprinosine 500 mg tablets—an immune system enhancer (4 tablets two times per day) for 15 days per month, for three consecutive months. Even though the opinions are contradictory, the systemic treatment was chosen according to the practitioner’s previous positive experience with this type of therapy in other HPV-related lesions or other diseases associated with a decrease in the immune response. The RCM examination (Figure 3, Figure 4 and Figure 5) was correlated with the histological findings and used in order to evaluate the treatment response. After 6 weeks of topic and systemic therapy, the evolution of the patient was favourable, with the disappearance of the majority of the lesions (Figure 1B). At the dermoscopy exam, we observed a reduction in the number of pigmented lesions and a decrease in the black-brown dots inside the pigmented regions. In addition, we observed the presence of an erythematous background due to the inflammatory effect of topical treatment with imiquimod 5% (Figure 1D). The RCM exam was also performed after six weeks of therapy, with a significant improvement in the architectural display of the cells, with the disappearance of acanthosis and papillomatosis and a considerable decrease in the number of melanophages (Figure 6). Taking into consideration the clinical, dermoscopic and confocal microscopy aspects of the lesion, it was decided to continue the application of topical treatment for three more weeks (3 applications/week), followed by a new RCM exam at the end of the therapy. The patient remains under follow-up.

## 3. Discussion

Over time, attempts have been made to define BP with therapeutic and prognostic implications. Summarising, BP represents a “high-grade squamous intraepithelial lesion” (HSIL) containing high-risk oncogenic HPV types or penile intraepithelial neoplasia III (PIN3), or histologically, a form of Bowen’s disease, but dependent on HPV status [9].

Dermoscopy in BP and penile intraepithelial neoplasia (PIN) has come into focus in the last few years, although the literature is limited regarding the characteristic aspects of these lesions [10].

The dermoscopy exam in BP is not pathognomonic. In most cases reported in the literature, pigment is observed in the form of grey-brown dots or granules with a linear disposition on a homogeneous brown background, dotted/glomerular vessels and the papillomatous surface of the lesion [10,11]. These findings can differentiate BP from other penile lesions, such as penile melanosis, seborrheic keratosis, and condyloma acuminatum. Still, dermoscopic differentiation from the other lesions classified as PIN is often a clinical challenge for the dermatologist.

Although clinically and prognostically different, the histopathological examination of BP highlights changes similar to Bowen’s disease: acanthosis with elongation of rete ridges, multiple mitoses, hyperchromatic nuclei, pleomorphic keratinocytes, multinucleated giant cells, and koilocytes, throughout the superior part of the epidermis, sometimes with perinuclear vacuolisation, without invasion of the basal membrane. Sometimes, the melanic pigment in increased quantity can be observed at the level of the lesions [12]. Pigmented lesions contain keratinocytes with increased melanin and melanophages in the upper dermis. In the superficial dermis, dilated and tortuous capillaries and perivascular inflammatory infiltrates of lymphocytes, plasmocytes, and histiocytes can be distinguished [13]. Differentiation from condyloma acuminatum is made by the presence of atypia and numerous mitoses up to the upper third of the epidermis, and in addition to Bowen’s disease, by the presence of koilocytes [2,13].

Although the clinical examination associated with dermatoscopy usually helps us establish a diagnosis with a fairly high accuracy, the potential for malignant transformation of this lesion requires a close follow-up of the evolution under treatment. Despite the small percentage of transformation of BP into SCC in situ (2.6%), this risk should not be neglected. Multiple high-risk HPV strains have been associated with malignant transformation of BP with genital localisation, including HPV-67 and HPV-31 strains [14]. The unpredictable evolution and genital location, which make it difficult to biopsy, require, when possible, to focus on a less invasive diagnostic and follow-up method, which also allows correlations with the initial histopathological exam [15].

Reflectance confocal microscopy (RCM) is one of these non-invasive imaging methods that allow us to evaluate genital pigmented lesions (diagnosis and assessment in dynamics of treatment response) by performing “virtual biopsies” with minimal discomfort for the patient [16]. Being a condition with a low frequency, the description of the RCM aspects is briefly presented in the literature, describing rather confocal aspects of Bowen’s Disease and Erythroplasia of Queyrat rather than BP. Thus, this case represents one of the few illustrations of RCM in bowenoid papulosis with genital localisation presented in the literature.

Regarding the RCM aspects, the literature reported the presence of dendritic cells at the level of the spinous and granular layers, dermal papillae surrounded by hyperreflective keratinocytes with cellular atypia, numerous small papillae with a hyperreflective outline and dilated blood vessels at the level of the superficial dermis [17].

The above-mentioned aspects were also found in our patient; RCM examination revealed pronounced acanthosis with the elongation of rete ridges, the presence of numerous dermal papillae, variable in size, with hyperreflective halo (edged papillae), centred by blood vessels, marked keratinocyte atypia and the presence of large, hyperreflective cells at the tip of the dermal papillae (Figure 3). The presence of these large, hyperreflective cells with star-shaped extensions (represented by melanophages) can cause difficulties in the differential diagnosis, with pigmented lesions being easily confused with the atypical melanocytes that bring to the fore the diagnosis of melanoma. However, although their star shape, large size (20–70 microns), and hyperreflective character are common to both cells, the differentiation by RCM can be done by visualising large, hyporefractile, eccentric nuclei in the case of atypical melanocytes, thereby allowing the differential diagnosis of BP and melanocytic pigmented lesions [18]. Although differentiating between melanocytes and melanophages can be easily done in RCM, a limitation of this method is the impossibility of distinguishing between the melanocytes and the Langerhans cells, both of them being nucleated, dendritic or round, hyperreflective, located in the suprabasal layer, immunohistochemistry studies on biopsy samples being necessary for this situation [19]. We also observe an alteration of the honeycomb pattern (atypical honeycomb pattern), translated histopathologically by the presence of numerous atypical cells, variable in shape and size at the level of the granular and spinous layers. They can also appear on RCM images with a targetoid aspect, a hyporeflective central nucleus and slightly hyperreflective surrounding cytoplasm, corresponding to dyskeratotic cells. We also observe an abundant vascularisation at the level of the dermal papillae with coiled, dilated vessels with mobile hyper-refractory figurative elements during real-time RCM examination.

The correlation of RCM findings with those encountered in the histopathological examination (Figure 2) allows us to follow in real-time the evolution of the lesions under treatment, as well as the potential transformation into SCC, without the need for additional biopsies. In addition, these correlations allow the clinician to establish the appropriate moment to stop the topical treatment, in which the benefits-risk ratio is highest, and the inflammatory process initiated by the imiquimod is still efficient in clearing the atypical cells.

## 4. Conclusions

Bowenoid papulosis is a relatively rare sexually transmitted disease classified as neoplasia in situ, with possible evolution to SCC. RCM is a non-invasive imaging technique that allows us to follow the evolution of the lesions without taking successive biopsies and without discomfort for the patient. The present article represents, to our knowledge, the first case of a correlation between RCM aspects and histopathological ones in BP. It is a starting point to highlight some RCM criteria that will allow minimally invasive diagnosis of BP in the future with high accuracy. Thus, the presence of marked acanthosis, with an atypical honeycomb pattern, melanophages and dendritic cells, although criteria are also present in Bowen’s Disease, allows correlating clinical and dermoscopic aspects to establish a non-invasive diagnosis by RCM. Moreover, the possibility of performing successive RCM exams allows the clinician to accurately establish the appropriate time for interrupting topical treatment. This is a method not only for diagnosis but also for treatment follow-up in order to obtain the best results and to reduce to a minimum the adverse effects of therapy.

## Figures and Tables

**Figure 1 diagnostics-13-01531-f001:**
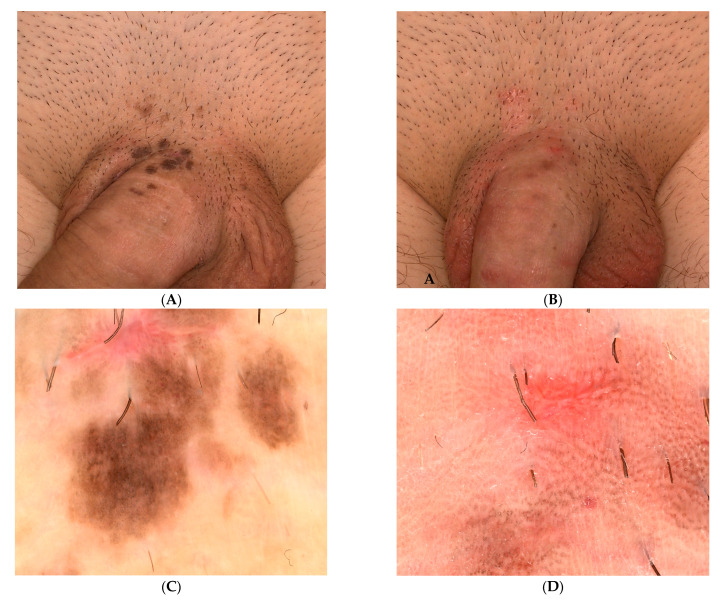
(**A**) Clinical aspect: multiple, well-defined, flat, pigmented violaceous papules with velvety surface at the root of the penis; (**B**)**.** Clinical aspects of the lesions at 6 weeks of topical treatment: a significant reduction in the number of the pigmented lesions; a decrease in pigmentation of the residual lesions; erythematous confluent patches at the root of the penis which extend towards the penis and scrotum due to application of Imiquimod 5%. (**C**)**.** Dermoscopy aspect of the lesions at the beginning of the treatment: pigmented papillomatous surface, brown-grey dots on a brown background, arranged in a linear pattern and widespread dotted vessels. On the upper part of the image, we can see an erythematous area corresponding to the biopsy scar. (**D**)**.** Dermoscopy aspect of the lesions at 6 weeks of topical treatment: reduction in the number of pigmentary lesions and black-grey dots; distribution of the remaining lesions on an erythematous background caused by the inflammatory reaction to Imiquimod application.

**Figure 2 diagnostics-13-01531-f002:**
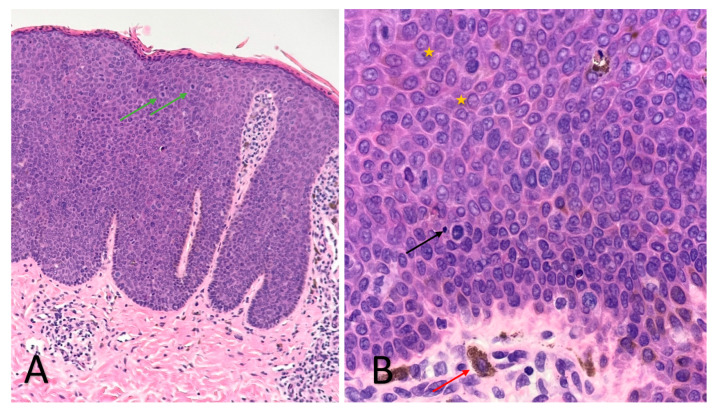
Histological aspects of the lesions of BP (haematoxylin-eosin stain): acanthosis with parakeratosis and lymphocytic infiltrate, with the preservation of the integrity of the basement membrane ((**A**) ×4); pleomorphic keratinocytes with atypia in the entire thickness of the epidermis (yellow star), with mitoses in the upper epidermis (green arrows), apoptotic cells with small pyknotic nuclei (black arrow) and melanophages (red arrow) ((**B**) ×40).

**Figure 3 diagnostics-13-01531-f003:**
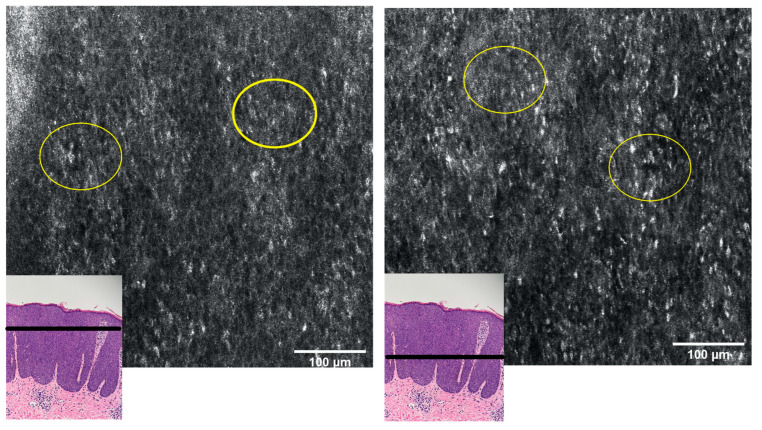
RCM mosaic at the level of spinous layer: alteration of the honeycomb pattern (atypical honeycomb pattern—yellow circle); cells with hyporeflective central nucleus and slightly hyperreflective surrounding cytoplasm, corresponding to dyskeratotic cells.

**Figure 4 diagnostics-13-01531-f004:**
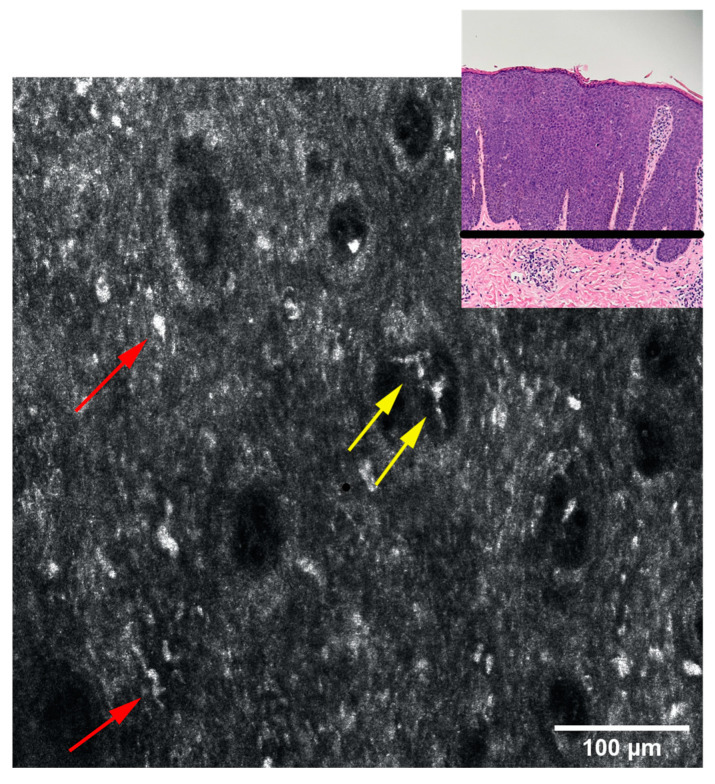
RCM mosaic at the level of the suprabasal epidermis: hyperreflective, elongated “pagetoid cells”—dendritic or round nucleated cells with visible nuclei, which histologically represent melanocytes or Langerhans cells (the differentiation cannot be achieved through RCM) (red arrows); melanophages inside dermal papillae (yellow arrows).

**Figure 5 diagnostics-13-01531-f005:**
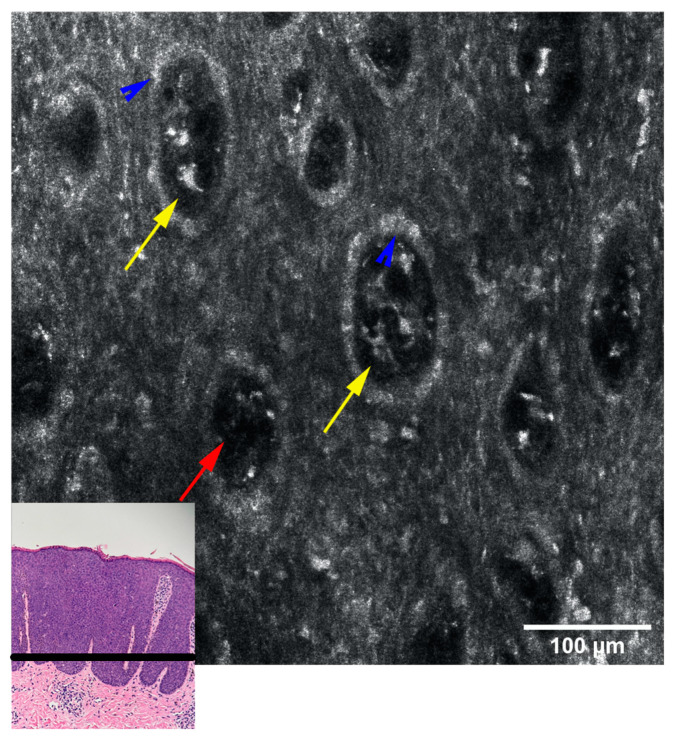
RCM mosaic at the level of dermo-epidermal junction: ringed pattern due to visualisation of dermal papillae with hyperreflective rings (blue arrow) formed by melanocytes and pigmented keratinocytes arranged at the basal level. Inside the dermal papillae large cells with irregular contours and imprecisely defined edges, with unidentifiable nuclei (“plump cells”) corresponding to melanophages (yellow arrow) and blood vessels (red arrow).

**Figure 6 diagnostics-13-01531-f006:**
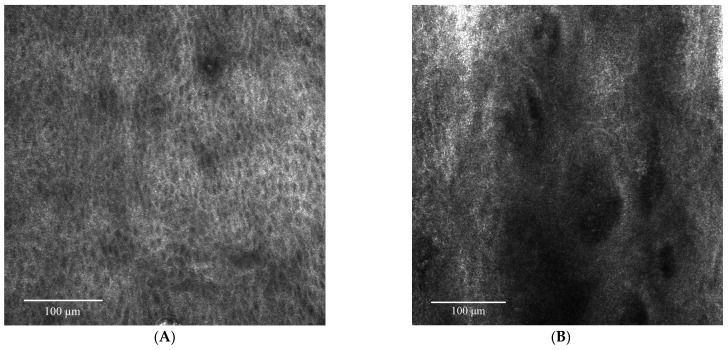
(**A**) RCM mosaic at the level of the granular layer: uniform honeycomb pattern, with a lower number of inflammatory cells (small bright cells) compared to the previous presentation due to the inflammatory process. (**B**) RCM mosaic at the level of dermo-epidermal junction: round ill-defined papillae with no large cells with irregular contours and imprecisely defined edges (that correspond to melanophages) as observed in the pre-treatment images.

## Data Availability

Not applicable.

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
