# Peer review of "Correlations between Histopathological and Confocal Reflectance Microscopy Aspects in a Patient with Bowenoid Papulosis"

_diagnostics, 2023, doi:10.3390/diagnostics13091531_

Round 1

Reviewer 1 Report

The authors present a nice case of genital Bowenoid papulosis treated with aldara and isoprinosine, monitored through RCM. Despite being clear and interesting as for the imaging modality, the authors should have discussed more in detail the pathogenic role of HPV in the presente case, therefore justifying systemic antiviral therapy.

Major points

- Immunohistochemical stain for p16 and HPV genotyping could have been performed to confirm the diagnosis. The authors should at least discuss the lack of such information, mentioning them in the discussion

- why wasn't a punch biopsy repeated at the end of the treatment to check the histological lack of bowenoid papulosis? this should be discussed.

Minor points

- Discussion is singular (chapter title)

- redundant terminology is often present and should be avoided ("manifested by the appearance”; "(for 6 weeks) [...]After 6 weeks")

- plural can often be avoided (e.g."repeated RCM examinations") 

- some repetitions are present ("is briefly presented in the literature, presenting rather confocal..")

Author Response

Thank you for your valuable comments. Please find attached the response letter.

Reviewer 2 Report

In this manuscript for a case report, a novel entity, reflectance confocal microscopy (RCM) of Bowenoid papulosis is presented. The topic of the manuscript is interesting, the clinical, histological and RCM images are of very high quality and aid the readers to appreciate the fine morphological details of this condition. In addition, the conclusions are supported by the reported case. However, there are some major and minor issues to be resolved by the authors:

Major issues:

1.      The title of the manuscript should be more concise, “Non-invasive diagnosis of genital lesions” is too general

2.      The abstract misses the conclusions

3.      The paper seems a bit too long, unrelated facts such as Mycoplasma infection should be omitted

4.      RCM images should be included in the case description, not in the discussion

5.      Isoprinosine is not a widely accepted evidence-based treatment of Bowenoid papulosis, this should be emphasized

Minor issues:

1.      In the abstract, I would not refer to Bowenoid papulosis as an STD

2.      Clinical and dermoscopy images should be put to the same figure

3.      Some RCM images lack scale bars

Author Response

Thank you for your comments. Please find attached the response letter. 

Round 2

Reviewer 2 Report

The paper has improved considerably and it is now suitable for publication.